# Geometry-Based Radiation Prediction of Laser Exposure Area for Laser Powder Bed Fusion Using Deep Learning

**Song Zhang \***, **Anne Jahn, Lucas Jauer and Johannes Henrich Schleifenbaum** 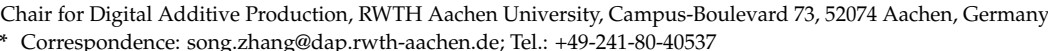

Chair for Digital Additive Production, RWTH Aachen University, Campus-Boulevard 73, 52074 Aachen, Germany
\* Correspondence: song.zhang@dap.rwth-aachen.de; Tel.: +49-241-80-40537

**Abstract:** Laser powder bed fusion (LPBF) is a promising technique used to manufacture complex geometries in a layer-wised manner. Radiation during the LPBF process is influenced by the part geometry, e.g., the overhang angle and the wall thickness. Locally varying radiation can cause deformation of the product after manufacturing. Thus, the prediction of the geometry-caused radiation before the manufacturing can support the evaluation of the design printability to achieve first-time-right printing. In this paper, we present a framework to predict the geometry-based radiation information using a deep learning (DL) algorithm based on the part geometry from computer-aided design (CAD). The algorithm was trained using data from an LPBF-print job consisting of parts with varying overhang angles. Image data, which include the information of radiation, were captured with an optical tomography (OT) camera system that was installed on a LPBF machine used in a laboratory environment. For the DL algorithm, a U-Net based network with mean absolute error (MAE) loss was applied. The training input was binarized OT data representing the contour of the designed geometry. Complementary, the OT data were used as ground truth for the model training. For the application, the design contours of multiple layers were extracted from the CAD file. The result shows the applicability to predict the OT-like radiation by its contour, which has the possibility to show the anomaly due to the part geometry.

**Keywords:** laser powder bed fusion; laser radiation prediction; deep learning; U-Net

## 1. Introduction

Metallic laser powder bed fusion (LPBF) additive manufacturing (AM) is a layer-based approach in which near-net shape parts are built directly from digital files [1]. In situ process monitoring sensors are used to obtain real-time information of the build process with the goal to connect in situ process signatures and post-process part quality [2]. Complex structures similar to overhangs imply manufacturing challenges due to the prevalence of distortion and dross defects. There is a large difference in the thermal conductivity of a solidified layer to the one of powder (powder only about 1%) [3,4]. Moreover, an unsupported overhang has no direct connection to the previously built layers or the build platform, and thus may be exposed to a lower cooling rate than the main structure. This can cause high thermal gradients resulting in differing material expansion behavior in areas of raised or reduced temperatures during the process of fast melting and solidification. This causes thermal stress and distortion [5–8]. These thermal behaviors were, for example, studied in simulation and experiment of forming T-shape overhangs [9]. Charles et al. [10], with varying process parameters, printed overhanging structures and examined the dimensional error statistically. They not only showed that the combined process parameters characterize the process, but especially that the energy density impacts the dross formation via an overheated overhang zone. Recently, Molnar et al. published a wide-ranging data set comparing measured and simulated temperatures in the fabrication of overhang structures. They also emphasized quality features such as microstructure, strain, and distortion directly related to the thermal history that the part was exposed to

and that accurate models of those phenomena likely depend on accurate, validated, thermal process models [11].

During the LPBF process, glow can be seen due to laser energy deposition at the current processing position, which contains the three components, namely reflected laser radiation, plasma emission, and temperature radiation. The optical tomography (OT)-based monitoring system was first introduced in [12] by filtering the reflection of laser radiation and plasma emission, which shows the possibility of monitoring temperature radiation from the building platform during the LPBF process over time. The captured radiation has positive correlation with melt pool temperature, according to the black body testing.

To monitor the thermal behaviors, the thermal modelling of the LPBF process can be classified into three categories: experimental measurement, numerical modeling, and analytical modeling. Experiment techniques apply sensors, e.g., pyrometer [13] and thermal camera [14], to measure the temperature during the process, which is inconvenient without manufacturing the parts. In the meantime, finite element analysis (FEM)-based numerical models and physical-based analytical models were introduced to predict temperature behavior. The FEM-based numerical models are reviewed in [15]. The main drawback of this is that, for large-scale parts, the computational time is high due to the iterative computation. Analytical models, which predict the temperature along the laser path without iterative computation, are introduced in [16,17]. However, the numerical models need the detailed material and process parameters for accurate prediction, which can be different for varied materials and machines.

Apart from numerical and analytical methods, machine learning (ML) methods enable handling AM process monitoring data with high dimensionality and complexity to predict the process behaviors in a data-driven manner. While there is an increase in available data in the AM lifecycle, Razvi et al. [18] state that there is still restricted knowledge in describing AM material–geometry–process–structure–property–performance relationships, and thus call for further research regarding computational and analytical tools.

Other work includes a prescriptive deviation-modelling method with ML techniques to model shape deviations in 3D printing. For a more precise tolerancing for AM parts, the geometric deviation patterns were estimated using Bayesian inference statistical learning from different shape data [18,19]. An overview of ML techniques for in situ process monitoring and control shows that most research has focused on analyzing process parameters (e.g., melt pool size) and detecting various defect types with visual, acoustic, or multi-sensor data [18]. Gobert [20] applied in situ OT data to a conditional generative adversarial network (cGAN) firstly, where the generator was able to imitate the images characteristics by part contour in current layer and scan vector orientation. However, the prediction of the OT-like images has no geometry dependency of printed parts, which can change the thermal behavior of the process and the method was not able to be transferred to large-scale parts due to its limitation of resolution of input images.

As laser energy emission and thermal conductivity during the manufacturing process impacts part quality, such as dross defects and distortion [4,18], this work not only focusses on a specific defect type, but uses radiation captured by an OT system and targets to predict the OT images from part geometry before manufacturing. This shows the potential to obtain energy-relevant behavior of the LPBF process before manufacturing. In this study, we present a method for OT-like radiation profile prediction by a U-Net structure using the contour information of a computer-aided design (CAD). This method shows the ability to predict the in situ radiation profile for large-scale parts before manufacturing and, in comparison to previous approaches, considers features such as overhangs using a multilayer input of a sliced part into the U-Net.

## 2. Materials and Methods

### 2.1. Methodology

In this section, a U-Net based framework (in Figure 1) is described for predicting the OT-like radiation distribution in the LPBF process using the contour information of a

printed part in a supervised learning manner. Firstly, a U-Net based network structure is described. Afterwards, the training procedure of proposed U-Net Model, as well as the application procedure, are explained separately.

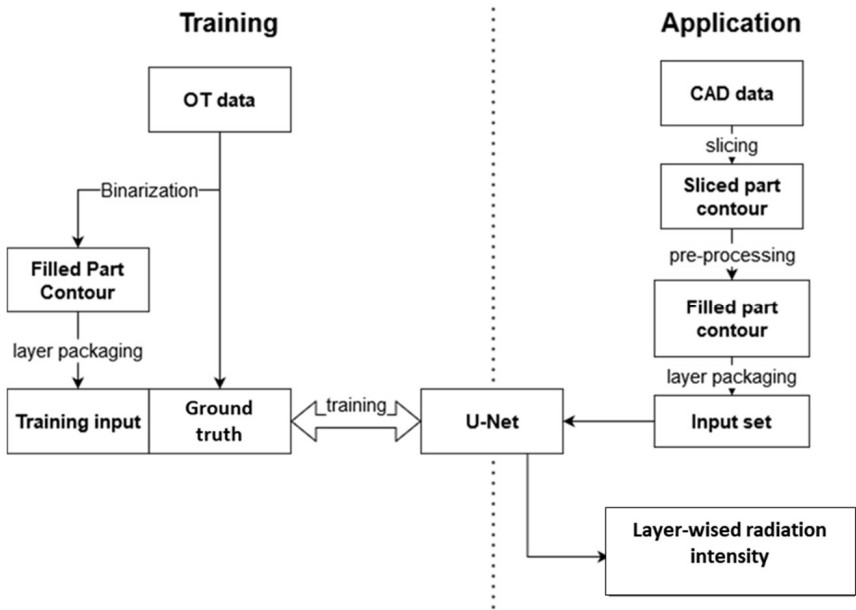

**Figure 1.** Framework for the prediction of the radiation.

The U-Net based network structure used in this work for predicting the radiation during the LPBF process is shown in Figure 2. This network structure is modified by the generator structure of Pix2Pix algorithm [21], which is widely used in the image process for image implanting and style transfer by using conditional generative adversarial network (cGAN). The application of the Pix2Pix algorithm in the AM field was introduced by Gobert [20], which showed the applicability of cGAN network for additive manufacturing to predict laser energy radiation. The U-Net-based generator in the Pix2Pix network has a depth of 8 for the input resolution of $256 \times 256$ pixels, where depth refers to the number of different spatially sized convolutional outputs.

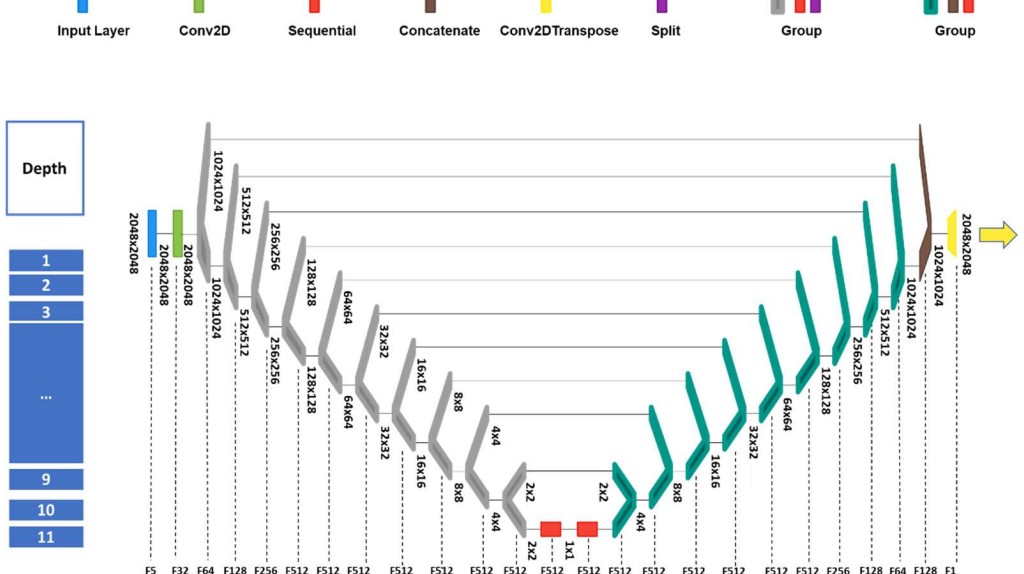

**Figure 2.** U-Net structure for energy distribution prediction via Net2Vis [22].

Due to varying part design, the size of printed parts may vary and thus input images need to depict the whole building platform. So, the input resolution of this network needs to be enlarged. According to the requirement of this network structure, the resolution of inputs needs to be $\left\{2^k \middle| k \in Z\right\}$. Thus, the input data have a size of [2048, 2048, 5]. This includes the adjusted image size with a length and width of each 2048-pixel for the current layer and 4 layers below, namely layer n, n − 1, n − 2, n − 3, and n − 4. The network's output is the predicted OT-like radiation of the current layer. To adapt the resolution, this U-Net adapts the depth to 11 pairs of downscaling and upscaling steps, while the mean absolute error (MAE) is used to train the model. The input data are scaled down 11 times to the resolution $1 \times 1$ at the bottleneck section and scaled up back to the output of $2048 \times 2048$ pixels. Conv2D and Conv2DTranspose layers use stride to downscale and upscale inputs by factor 2. The sequential layer for downscaling contains a Conv2D layer with activation function ReLU, while the upscaling layer contains a Conv2DTranspose layer with activation function LeakyReLU. The pairs of downscaling and upscaling layers with identical size are concatenated to build up the U-shape of the network. The other hyperparameters remain the same in the original generator for the Pix2Pix algorithm. The resolution and filter amount at each layer are marked in Figure 2. To evaluate the network performance with varied depth, a comparison between networks with depths 11, 9, 7, 5, and 3 is applied and shown in Section 3. In the meantime, this proposed network structure will be compared with the Pix2Pix algorithm in Section 3.

The training dataset is constructed from the monitoring data of an OT monitoring system, which is introduced in Section 2.2. The data are pre-processed to present the contour information of the printed parts via binarization and layer packaging.

Considering mesh-based CAD files, it is difficult to use these designs to obtain the part contour as it lacks information of the part location on the building platform. Thus, the mesh-based design cannot be aligned and used with the bitmap-based monitoring data. On monitoring images, the laser exposure area has a high contrast to the background area (in Figure 3a). These areas can be binarized using the Ostu binarization algorithm [23], according to a histogram (in Figure 3b). The binarized monitoring data are shown in Figure 3c using a grey value of 5000 as the threshold, which show the separation of the background and laser exposure area.

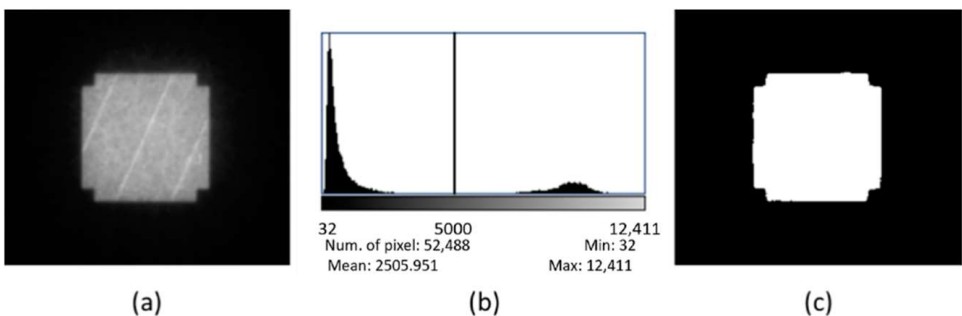

**Figure 3.** (**a**) Monitoring data of a part; (**b**) histogram of grey value of the monitoring data; (**c**) binarized monitoring data using 5000 as the threshold.

To represent part geometry, 5 layers are packed together as input for the network. To cover a wider range in build direction, these layers consist of the current layer n and 4 previous layers, namely layer n − 1, n − 2, n − 3, and n − 4. This allows the network to distinguish the different geometries (e.g., the overhang angle).

Normalization is applied to the training data set to achieve better coverage during network training. The 16-bit grey value of the ground truth and the predictions are normalized from $[0, 2^{16} - 1]$ to $[-1, 1]$, where

$$\text{normalized grey value} = \frac{2 \times \text{ grey value}}{2^{16} - 1} - 1 \tag{1}$$

Considering the stochasticity of the LPBF process, the energy radiation can be different even with identical geometry and parameter configuration. It can create critical outlier values regarding the geometries (e.g., overhang angles). Since the training procedure should not be led by this stochasticity, the outlier values should not be emphasized during training. Therefore, the mean absolute error (MAE) is used to train the model as loss function instead of the mean square error (MSE) because all errors share the same weight. The MAE loss function can be represented as follows:

$$L_{MAE}(\hat{y}, y) = \frac{1}{M} \sum_{i=1}^{M} |\hat{y}_i - y_i| \tag{2}$$

where $M$ is the total pixel amount of an image, while $\hat{y}$. and $y$ are images with predicted radiation profile and real OT images. A lower MAE loss indicates higher prediction accuracy. All training procedures are applied on a Nvidia DGX workstation with 4 V100 GPUs and use an Adam optimizer with a learning rate of 0.0002.

For a newly designed part, unlike in the training procedure, the contour is acquired directly from its CAD file. Considering that the spatial resolution of the OT camera is fixed, the resolution of a bitmap can be aligned using digit per inch (DPI) ratio to adapt the physical size. The mesh-based CAD file will be sliced, according to pre-defined layer thickness and orientation, into images based on the pre-defined DPI. After obtaining the contour of the part, to construct the input data for the U-Net network, the adapted sliced bitmaps are packed using layer packaging.

## 2.2. Experiment

For the in situ monitoring of the LPBF process, an OT system is installed on EOS M 290, which uses a sCMOS (scientific complementary metal-oxide semiconductor) camera PCO.edge 5.5 and works in a near-infrared (NIR) spectral range. The field of the camera view covers the whole building platform with a size of 250 × 250 mm. Additional technical information of the camera is provided in Table 1.

**Table 1.** Technical specifications of the sCMOS camera (h: horizontal, v: vertical).

| Camera Feature | Specification |
| --- | --- |
| Type of sensor | sCMOS monochrome |
| Bit depth | 16 bits |
| Resolution (h × v) | 2560 × 2160 active pixel |
| Pixel size (h × v) | 6.5 μm × 6.5 μm |
| Exposure time | 2 s rolling shutter |

Radiation during the LPBF process contains the process quality information since it is a representative of the emission energy of the melt pool where the metallic powder melts and consolidates. The monitoring system should enable the capturing of the radiation layer-by-layer during the LPBF process. Thus, a bandpass filter with a 100 nm width and a center wavelength (CWL) of 900 nm is used to block the laser reflection and avoid the saturation of the camera sensor due to high-intensity inputs in the broad spectrum.

Monitoring data are acquired continually at each layer during the LPBF process. Since exposure time of the camera is not long enough to capture the complete laser exposure process of a layer, multiple images are taken for each layer with a chosen exposure time of 2 s. All images of a certain layer are then stacked to a single image by summing up all grey values pixel-wise.

The acquired OT images are calibrated according to the lookup tables (LUTs) for flat-field, defective pixel, and perspective correction where the LUTs are obtained from camera calibration experiments. These calibrated OT images cover the whole 250 × 250 mm building platform of the EOS M290 with a resolution of 2125 × 2125 pixels. The corresponding spatial resolution of the OT monitoring system is 250 mm/2125 pixel = 0.118 mm/pixel.

The process parameter configuration used is shown in Table 2. The corresponding volume energy density is $E_V = \frac{P_L}{d_s v_s \Delta y_s} = 62.5 \, \text{J/mm}^3$ [24]. As a rule of thumb, overhang angles less than 45° can be printed using additive manufacturing without support structure. To increase the probability of encountering anomalies in radiation, 15 parts with different overhang angles between 45° to 90° with an interval of 5° are printed for model training (in Figure 4a). The parts are located on the building platform randomly (Figure 4b). Each designed part consists of a strut section and overhang angle section (in Figure 4c,d). The height of strut sections and overhang sections of each part are different to reduce the coupling in training data. To avoid critical geometric distortion (e.g., warping), which may stop the process, parts with overhang angles 75°, 80°, 85° and 90° are cancelled manually after the manufacturing of several overhang layers (in Table 3). Sandvik 17-4PH powder with a mean particle size 6.7 μm is used for the print job. To create training data with different behaviors of the printing process, struts and overhangs have varying heights. The print job contains 662 layers, while 652 pairs of data are generated for the training due to layer packaging. All captured monitoring data with the process are presented in Section 2.1.

**Table 2.** Job property of training set generation.

| Feature | Value |
| --- | --- |
| Layer thickness $D_s$ | 0.03 mm |
| Laser power $P_L$ | 120 W |
| Hatch distance $\Delta y_s$ | 0.08 mm |
| Scanning speed $v_s$ | 800 mm/s |

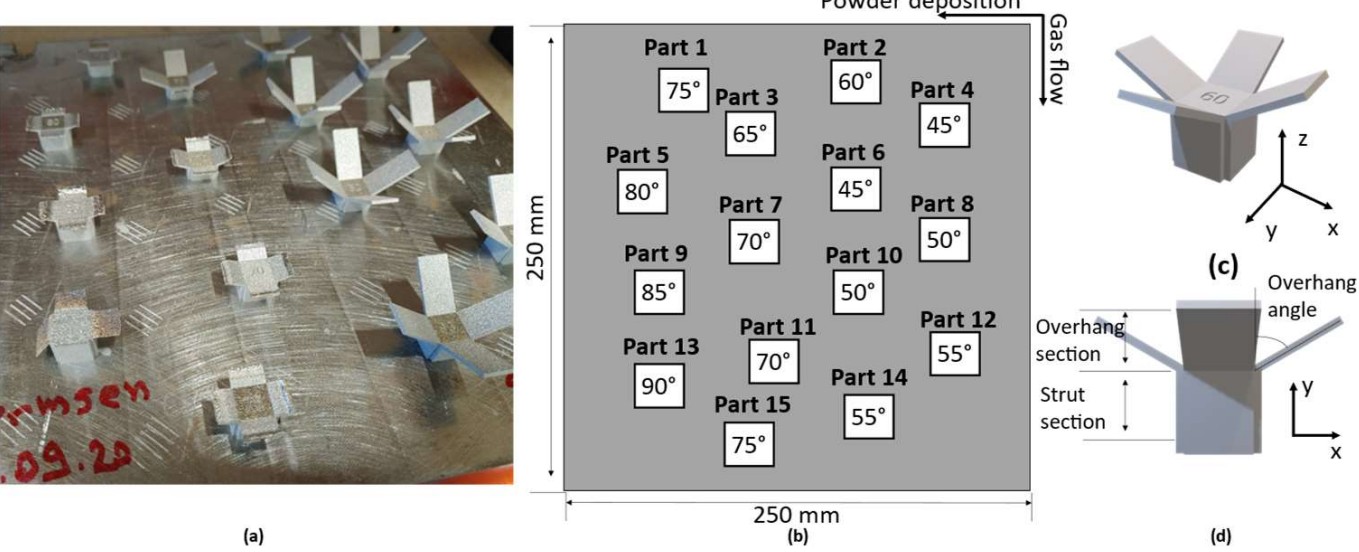

**Figure 4.** (**a**) Printed parts for training data set; (**b**) orientation of printed part on building platform of EOS M290; (**c**) design of part 2 with 60° overhang angle; (**d**) front view of designed part 2.

The test part is printed with the identical LPBF parameter configuration as the training set. The designed CAD file in STL is sliced by the Trimesh library in python and packaged according to the application procedure in Section 2.1. The geometry of test design is shown in Figure 5. The part is sliced by 500 layers, each with layer thickness of 30 μm, and printed in the center of 250 × 250 mm building platform. The layer-wised monitoring data from the OT camera is used as ground truth to evaluate the performance. The sliced STL file is placed in the center of the 2048 × 2048 image, which is aligned with monitoring data. The test geometry contains the hold structures, which provide the overhang structures for critical radiation.

**Table 3.** Layer height of printed parts.

| Part | Overhang Angle | Layer Amount of Strut Section | Layer Amount of Printed Overhang Section |
|------|----------------|------------------------------|------------------------------------------|
| 1 | 75° | 257 | 21 |
| 2 | 60° | 157 | 242 |
| 3 | 65° | 202 | 197 |
| 4 | 45° | 154 | 511 |
| 5 | 80° | 340 | 3 |
| 6 | 45° | 160 | 495 |
| 7 | 70° | 207 | 53 |
| 8 | 50° | 239 | 426 |
| 9 | 85° | 341 | 17 |
| 10 | 50° | 234 | 431 |
| 11 | 70° | 279 | 48 |
| 12 | 55° | 307 | 358 |
| 13 | 90° | 366 | 1 |
| 14 | 55° | 304 | 361 |
| 15 | 75° | 310 | 20 |

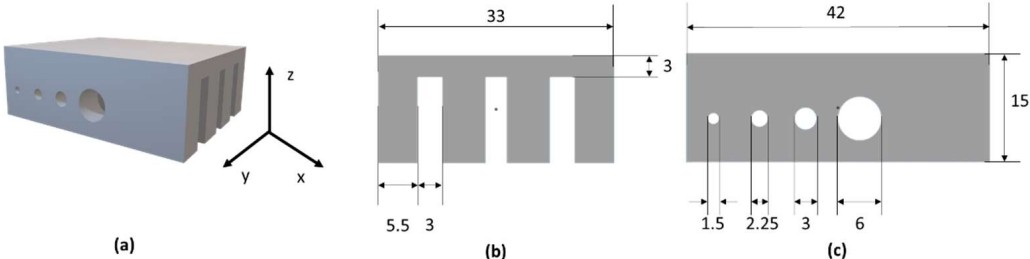

**Figure 5.** (**a**) Test job; (**b**) front view of the test job; (**c**) left view of the test job.

## 3. Results

The MAE loss out of training process reached 0.000826 after 100 epochs, while for each layer, the average radiation was predicted every 0.54 s with a V100 GPU. Since the pixel value was normalized during training, the pixel-wised MAE of grey value was $\frac{0.000826}{2} \times 2^{16} = 27.07$ in 16-bit image. For the training job, the mean grey value through the training data set was 320.07, so the average of pixel-wised prediction error was $\frac{27.07}{320.07} \times 100\% = 8.46\%$. Since the MAE loss was calculated through the calibrated OT image of whole building platform with a resolution of 2125 × 2125 pixels, the prediction error was influenced by the part amount and orientation on the building platform.

Considering the network performance of a different depth, the comparison result is shown in Figure 6. All networks were trained for 100 epochs and tested by the test set. The high depth benefits the network performance and leads to low MAE loss. The network with a depth of 11 shows the lowest MAE in both the training and test set. Since the training and test dataset were not relevant and there was only one part on the building platform in the test dataset, the test loss can be lower than the training loss due to the average of MAE over the whole building platform. The result showed that a high depth of UNet structure provides more tunable parameters under lower resolutions due to the downscaling and upscaling behavior, which was beneficial to the radiation prediction.

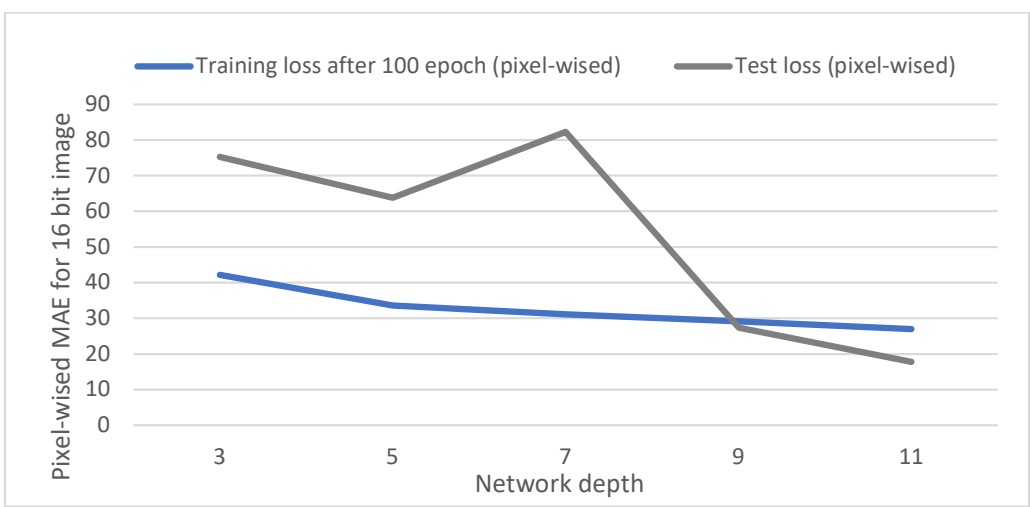

**Figure 6.** Network performance with different depths.

The comparison of results shows test losses in both networks, and the standard deviation of test errors in Figure 7. Pix2Pix and U-Net show a similar performance in MAE, while the Pix2Pix algorithm shows a higher standard deviation of test error.

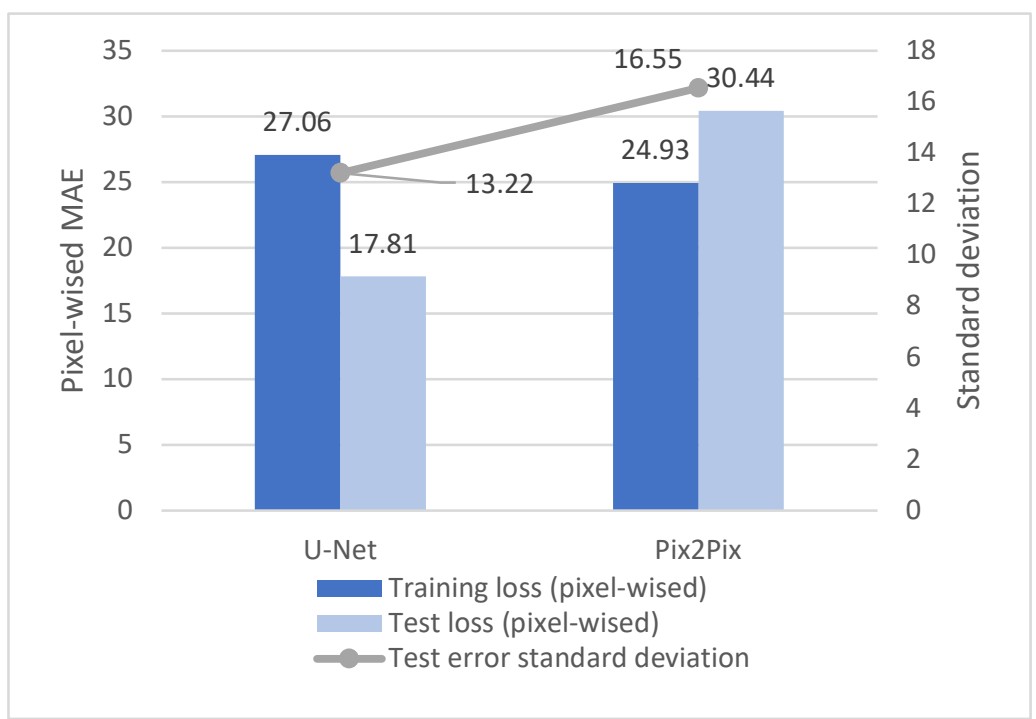

**Figure 7.** Comparison of the performances of Pix2Pix and U-Net.

The test results of the presented U-Net based framework and Pix2Pix structure are shown in Figure 8. To visualize the radiation, the grey value was mapped using a look-up table (LUT) of 'thermal' in Fiji [25]. High radiation on layer 300, due to the critical overhang structure on the top of the hole, was predicted by both algorithms.

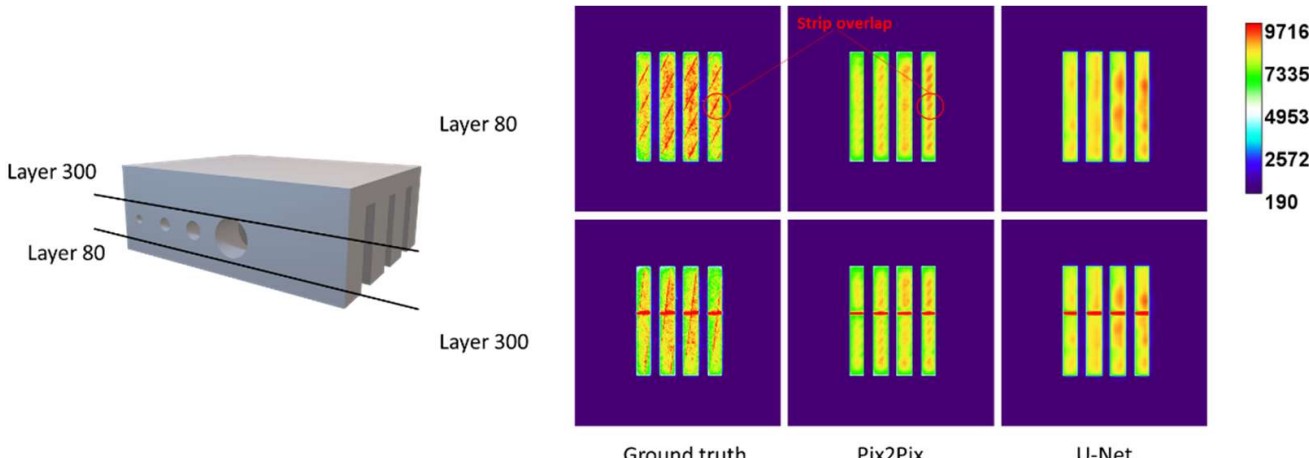

**Figure 8.** Test results of layer 80 and layer 300 for different network structures.

## 4. Discussion

The OT camera system has been proved to be sufficient to capture the radiation layer-wise during the manufacturing process. This provides the possibility of understanding energy-relevant behavior of the LPBF process, which can link to the product quality in the end of the process chain. Unlike the existing research in [1], the proposed method focused on the radiation prediction according to the geometry of the designed part using its sliced contour. The result shows the capability of predicting the critical anomaly of radiation. Since the inputs are only contours of a designed part in slice, the strip overlap due to scan vector orientation should not be predictable. The Pix2Pix algorithm made a guess of possible strip overlap due its generative structure by the discriminator of GAN. In this case, the guess is not plausible, according to ground truth. Considering that the wrongly predicted strip area has higher radiation than its surroundings, this can be classified as anomaly radiation. Due to this potentially wrong classification, U-Net is more stable than the Pix2Pix algorithm. This method enables the investigation of energy-relevant behaviors without real processing or simulation, which provides the possibility of evaluating and optimizing the part orientation in data preparation. However, on the one hand, the prediction considers only geometry and is not yet fully fit to the real case due to the lack of information on the process, for example, scan vector orientation and laser parameter. On the other hand, the predictive model can only cover common cases in the manufacturing process. The stochasticity from the machine, powder bed, and environment, which will occur during the LPBF process, cannot be predicted. This will lead to potential defects in the final product.

In the meantime, the training dataset was captured and processed by manufacturing parts with overhang angles. Radiation on the geometries with overhang angles in the test dataset was predicted due to the availability of data with similar geometries. In Figure 9, the prediction results for layer 80 and 300 of the test dataset using trained model via the full dataset and data from the first 150 layers are shown. Since the first 150 layers in the training dataset contains a strut section with an identical size for each part (see in Table 3), radiation in the overhang section at layer 300, as well as the inner section at layer 80, was predicted with offset by the prediction model using the first 150 layers. In this regard, the shape size of the designed geometry for building up training sets can influence the accuracy of the prediction model. Furthermore, since the radiation during the LPBF process is varied for different materials and different laser parameters, the prediction model is applicable under the configuration, which is identical to the training set. For new configurations, the model needs to be trained with appropriate data.

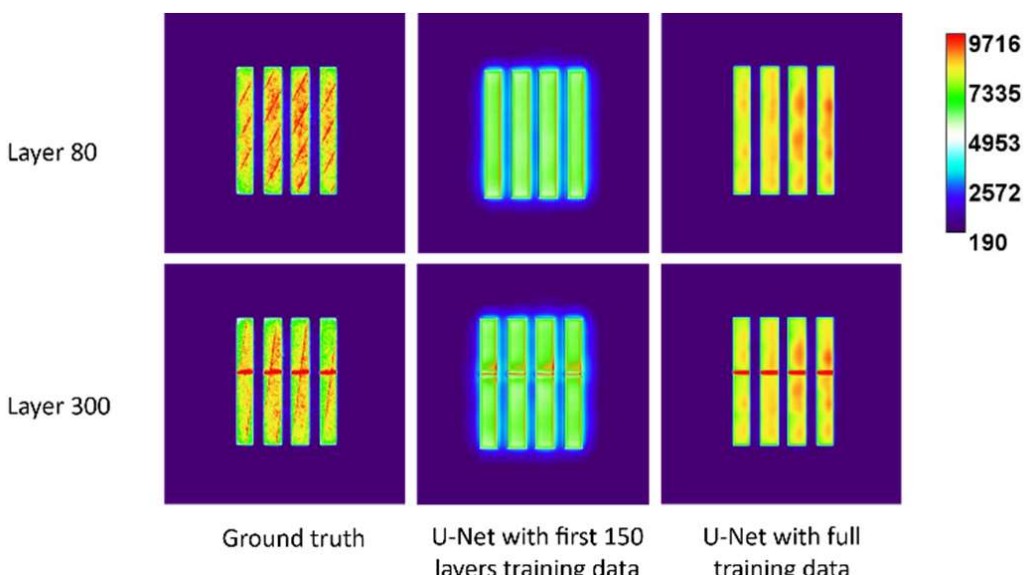

**Figure 9.** Prediction results by training data of the first 150 layers and full training data at layers 80 and 300.

## 5. Conclusions

In this study, a U-Net based framework for the radiation prediction of the LPBF process was introduced, which includes data pre-processing, training, and application of the prediction model. Training and test datasets were built using an integrated OT monitoring system. To represent the geometry of the printed parts, contours of five layers were used as input data. In addition, to enable the prediction of radiation for large-scaled parts, the resolution of input of the network was adapted to cover the whole building platform. The results showed that the proposed framework indicates its capability to predict radiation layer-wise by the contour of the designed part. This framework provides a data-driven method, besides numerical and analytical methods, to obtain a fast result of radiation, which can be used as a quick check of thermal behavior before the real manufacturing.

For the future work, the prediction accuracy needs to be improved. Since the current input is the contour of the designed part, other types of information, which can influence the radiation, including scan vector orientation and laser parameter configuration, should be investigated to have a more complete model. Meanwhile, different material needs to be discussed as well. Furthermore, hyperparameter tuning will be of benefit to reduce the prediction error.

**Author Contributions:** Conceptualization, S.Z. and A.J.; methodology, S.Z.; software, S.Z.; validation, A.J. and L.J.; formal analysis, A.J. and L.J.; investigation, S.Z.; resources, S.Z.; data curation, S.Z.; writing—original draft preparation, S.Z.; writing—review and editing, A.J. and L.J.; visualization, S.Z. and A.J.; supervision, J.H.S.; project administration, J.H.S.; funding acquisition, J.H.S. All authors have read and agreed to the published version of the manuscript.

**Funding:** This research was funded by the Deutsche Forschungsgemeinschaft (DFG, German Research Foundation) under Germany 's Excellence Strategy–EXC-2023 Internet of Production–390621612.

**Institutional Review Board Statement:** Not applicable.

**Informed Consent Statement:** Not applicable.

**Data Availability Statement:** The data presented in this study are available on request from the corresponding author. The raw/processed data needed to reproduce these findings cannot be shared publicly at this time, as they are also part of an ongoing study.

**Conflicts of Interest:** The authors declare no conflict of interest.

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
