# Peer review of "Geometry-Based Radiation Prediction of Laser Exposure Area for Laser Powder Bed Fusion Using Deep Learning"

_applsci, doi:10.3390/app12178854_

Round 1

Reviewer 1 Report

This work presented a deep learning-based prediction model for the geometry-related radiation in the laser powder bed fusion. Overall, the topic is worthy of investigation and within the scope of the journal. It can be accepted for publication after fully addressing the following issues.

1)      The literature review is inadequate without discussing the physics behind the radiation during LPBF and recent advanced physics-based modeling works. Please review the following references for more information. doi.org/10.3390/ma13081988: etc.

2)      The gap between the current work and existing works should be further discussed.

3)      The influence of the data availability in terms of size and accuracy on prediction accuracy should be further discussed, which could lead to the drawback in terms of predictability with process variation and material difference.

4)      The computational accuracy should be discussed for readers to understand the applicability of the radiation prediction of large-scale parts in LPBF.

5)      The details of the deep learning model should be elaborated.

6)      The reference links are broken and needed to be fixed in the revision.

7)      The conclusion should include the key findings, the contribution, and the significance of the current work. 

Reviewer 2 Report

The authors presented a framework to predict the geometry-based radiation information using a deep-learning algorithm based on the CAD part geometry. Their results showed the applicability to predict the OT-like radiation by its contour, which has the possibility to show the anomaly due to the part geometry. The study is good, but some questions need to be clarified.

1.       The word "error! Reference source not found.." appears many times in the manuscript, and the author should carefully check it in the full text.

2.       Please write the full name of the abbreviation "CAD" in the manuscript.

3.       In line 187 of the article, "parts with high overhang angle were cancel manually after manufacturing of several overhang layers (in Table 3) ". How to define the high overhang angle? or which angles are high overhang angles?

4.       In Table 3, why does part 1 2 3 4 5 appear twice? What's the difference between them?

5.       Figure 6 shows that the network with depth of 11 shows the lowest MAE in both training and test set. Can the author give a reasonable explanation for this phenomenon?

6.       The format of the references in the manuscript should be consistent.

Round 2

Reviewer 1 Report

The authors have addressed the raised issues in the revised manuscript. The reviewer has no further concerns regarding the current work. It can now be accepted for publication. 

Reviewer 2 Report

The authors responded positively. I think the manuscript can be published in the present form.